# Bile Acid Metabolism Analysis Provides Insights into Vascular Endothelial Injury in Salt-Sensitive Hypertensive Rats

**DOI:** 10.3390/metabo14080452

**Published:** 2024-08-16

**Authors:** Baihan Zeng, Xile Peng, Li Chen, Jiao Liu, Lina Xia

**Affiliations:** School of Health Preservation and Rehabilitation, Chengdu University of Traditional Chinese Medicine, Chengdu 611137, China; zbh@stu.cdutcm.edu.cn (B.Z.); peng_xile@163.com (X.P.); chenli_cdutcm@163.com (L.C.); liujiao@cdutcm.edu.cn (J.L.)

**Keywords:** vascular endothelial injury, hypertension, high salts, bile acid, metabolomics

## Abstract

As an unhealthy dietary habit, a high-salt diet can affect the body’s endocrine system and metabolic processes. As one of the most important metabolites, bile acids can prevent atherosclerosis and reduce the risk of developing cardiovascular diseases. Therefore, in the present study, we aimed to reveal the bile acid metabolism changes in salt-sensitive hypertension-induced vascular endothelial injury. The model was established using a high-salt diet, and the success of this procedure was confirmed by detecting the levels of the blood pressure, vascular regulatory factors, and inflammatory factors. An evaluation of the histological sections of arterial blood vessels and kidneys confirmed the pathological processes in these tissues of experimental rats. Bile acid metabolism analysis was performed to identify differential bile acids between the low-salt diet group and the high-salt diet group. The results indicated that the high-salt diet led to a significant increase in blood pressure and the levels of endothelin-1 (ET-1) and tumor necrosis factor-α (TNF-α). The high-salt diet causes disorders in bile acid metabolism. The levels of four differential bile acids (glycocholic acid, taurolithocholic acid, tauroursodeoxycholic acid, and glycolithocholic acid) significantly increased in the high-salt group. Further correlation analysis indicated that the levels of ET-1 and TNF-α were positively correlated with these differential bile acid levels. This study provides new evidence for salt-sensitive cardiovascular diseases and metabolic changes caused by a high-salt diet in rats.

## 1. Introduction

Salt is a common condiment in daily life, and has anti-inflammatory and bactericidal effects [1]. However, the intake of salt is not necessarily a case of the more, the better. The World Health Organization recommends that each person consume no more than 6 g of salt per day [2]. Excessive salt intake can increase the osmotic pressure of the plasma and the burden on the kidneys [3], and it can also damage the gastric mucosa, increasing the risk of developing gastric cancer [4]. Several studies have demonstrated that many people consume excessive salt in their diet [5,6]. This indicates that a high-salt diet has become a health hazard, and we assert that it is one that cannot be ignored.

Cardiovascular diseases (CVDs) are among those with the highest incidence rates in the world, including hypertension, coronary heart disease, vascular damage, etc. A high salt intake is an important cause of CVDs. A 2023 research survey showed that a high-salt diet increases the risk of developing CVDs and thus highlighted it as a potential socio-economic issue [7]. Vinaiphat’s research confirms that high-salt-induced hypertension can cause vascular endothelial damage [8]. Therefore, to enable us to better prevent and control CVDs, it seemed that a more in-depth study should be conducted on the correlation between salt intake and CVDs.

Bile acids are the main component of human bile and the final product of cholesterol after liver action. Bile acids are widely present in the plasma and regulate a variety of physiological functions through enterohepatic circulation, such as reducing blood fat, which can prevent atherosclerosis and protect blood vessels [9,10]. As an important component of the metabolic process and endocrine system, bile acids are thought to have an impact on the occurrence and development of CVDs [11,12]. Ishimwe et al. have discussed the association between bile acids and salt-sensitive hypertension [13]. Therefore, we attempted to identify the changes in bile acid metabolism under a high-salt diet, and we explored a new idea for preventing and treating salt-sensitivity-induced CVDs.

This study used bile acid metabolism analysis to gain insights into vascular endothelial injury in salt-sensitive hypertensive rats. We detected blood pressure, vascular regulatory factors, and inflammatory factors to clarify whether a high-salt diet can cause salt-sensitive hypertension and vascular endothelial injury. Furthermore, we used bile acid metabolomics analysis to identify the changes in bile acids. We identified bile acids closely related to vascular endothelial injury through correlation analysis. In general, this approach explores the effects of high salt at the metabolic level and its correlation with CVDs. The main schematic flowchart of this study is shown in Figure 1.

## 2. Materials and Methods

### 2.1. Animals and Diet

Dahl salt-sensitive (DS) rats and Wistar rats were purchased from SiPeiFu Biotechnology Co., Ltd. (Beijing, China). The rat feeds were purchased from KeaoXieli Feed Co., Ltd. (Beijing, China). We housed 12 male DS rats and 6 male Wistar rats (200 ± 20 g) in a well-ventilated room at room temperature, with a 12 h light–dark cycle and free access to food and pure water. After one week of adaptive feeding, we randomly assigned the DS rats to two groups: the low-salt-diet group (LSD group, 0.3% NaCl diet feeding) and the high-salt-diet group (HSD group, 8% NaCl diet feeding). The Wistar rats were set as the normal control group (NC group, 0.3% NaCl diet feeding). The catalog number for the 0.3% NaCl diet is 1016712327837081600, and that for the 8% NaCl diet is 1016706714625204224. Detailed composition information is shown in Appendix A.

### 2.2. Sample Collection

At the end of adaptive feeding, and the 8th week after modeling, the blood pressure of the rats of three groups was measured by the non-invasive blood pressure analyzer (BP-2010A). Afterward, the rats were anesthetized with 1% pentobarbital sodium (30 mg/kg) and sacrificed. Blood samples of ~10 mL were collected from each rat’s abdominal aorta. The blood samples were centrifuged for 10 min at 3000 r/min at 4 °C to obtain plasma samples. The levels of plasma endothelin-1 (ET-1), nitric oxide (NO), Angiotensin-II (Ang-ΙΙ), tumor necrosis factor-α (TNF-α), interleukin-6 (IL-6), and interleukin-10 (IL-10) were determined by enzyme-linked immunosorbent assay (Elisa) kits from Elabscience Biotechnology Co., Ltd. (Wuhan, China). The remaining plasma samples were used for bile acid metabolism analysis. The abdominal aorta and renal tissue samples of rats were obtained for histopathological analysis. All samples were stored at −80 °C until use.

### 2.3. Vascular and Renal Histopathological Examination

The aorta and renal tissue samples were fixed with 10% neutral formaldehyde for 48 h and cut into 5 mm sections. Paraffin sections were prepared and stained with hematoxylin–eosin (H&E). The vascular endothelial injury of sections was observed and photographed using an optical microscope with a magnification of 40×, 100×, and 200×. The renal injury of sections was observed and photographed using an optical microscope with a magnification of 100× and 200×.

### 2.4. Bile Acid Metabolism Analysis

#### 2.4.1. Metabolite Extraction

A 100 μL plasma sample was transferred to an Eppendorf tube. After the addition of 400 μL of extract solvent (acetonitrile–methanol, 1:1, precooled at −40 °C), the samples were vortexed for 30 s, sonicated for 10 min in an ice water bath followed by incubation at −40 °C for 1 h and centrifugation at 12,000 rpm (RCF = 13,800× *g*, R = 8.6 cm) and 4 °C for 15 min. The clear supernatants were transferred to an auto-sampler vial for ultra-high-performance liquid chromatography–parallel reaction monitoring–mass spectrometry (UHPLC-PRM-MS) analysis.

#### 2.4.2. Standard Solution Preparation

Stock solutions were individually prepared by dissolving or diluting each standard substance to give a final concentration of 1 mg/mL. An aliquot of each stock solution was transferred to a flask to form a mixed working standard solution. A series of calibration standard solutions were then prepared by stepwise dilution of this mixed standard solution (containing the isotopically labeled internal standard mixture in concentrations identical to the samples).

#### 2.4.3. UHPLC-PRM-MS Analysis

The UHPLC separation was carried out using a UHPLC System (Vanquish, Thermo Fisher Scientific, Waltham, MA, USA), equipped with a Waters ACQUITY UPLC BEH C18 column (150 × 2.1 mm, 1.7 μm, Waters, Milford, MA, USA). Mobile phase A was 5 mmol/L ammonium acetate in water, and phase B was acetonitrile. The column temperature was set at 45 °C. The auto-sampler temperature was set at 4 °C and the injection volume was 1 μL.

An Orbitrap Exploris 120 mass spectrometer (Thermo Fisher Scientific) was applied for assay development. Typical ion source parameters were spray voltage = +3500/−3200 V, sheath gas (N_2_) flow rate = 40, aux gas (N_2_) flow rate = 15, sweep gas (N_2_) flow rate = 0, aux gas (N_2_) temperature = 350 °C, capillary temperature = 320 °C.

The parallel reaction monitoring parameters for each of the targeted analytes were optimized by injecting the standard solutions of the individual analytes into the application programming interface source of the mass spectrometer. Since most of the analytes did not show a product ion acceptable for quantification, the precursor ion in high resolution was selected for quantification.

#### 2.4.4. Calibration Curves

Calibration solutions were subjected to UHPLC-PRM-MS/MS analysis using the methods described above. The least squares method was used for the regression fitting, and 1/x weighting was applied in the curve fitting since it provided the highest accuracy and correlation coefficient (R^2^). The level was excluded from the calibration if the accuracy of the calibration was not within 80–120%.

#### 2.4.5. Limit of Detection and Limit of Quantitation

The calibration standard solution was diluted stepwise, with a dilution factor of 2. These standard solutions were subjected to UHPLC-PRM-MS analysis. The signal-to-noise ratios (S/N) were used to determine the lower limits of detection (LLODs) and lower limits of quantitation (LLOQs). The LLODs and LLOQs were defined as the analyte concentrations that led to peaks with signal-to-noise ratios (S/N) of 3 and 10, respectively, according to the US FDA guidelines for bioanalytical method validation.

#### 2.4.6. Precision and Accuracy

The precision of the quantitation was measured as the relative standard deviation (RSD), determined by injecting analytical replicates of a quality control (QC) sample. The accuracy of quantitation, measured as the analytical recovery of the QC sample, was determined. The percent recovery was calculated as [(mean observed concentration)/(spiked concentration)] × 100%.

#### 2.4.7. The Detection Result of Bile Acids

Appendix A shows the extracted ion chromatographs from a standard solution (Appendix A) and a sample (Appendix A) of the targeted analytes under optimal conditions. As can be seen from the figure, (i) most of the analytes showed excellent peak shapes, and (ii) good separations were obtained. In samples, a total of 38 target metabolites were detected. After collecting bile acid data, enrichment analysis and plotting were completed by MetaboAnalyst 5.0.

### 2.5. Correlation Analysis

After obtaining vascular and metabolomics data, we used the ‘Wu Kong’ platform for correlation analysis, available at https://www.omicsolution.com/wkomics/main/ (accessed 1 May 2023).

### 2.6. Statistical Analysis

For statistical analysis, values were expressed as means ± standard deviation. Data were analyzed using SPSS (version 26.0) statistical software. Statistical significance among groups was determined using the independent *t*-test followed by post hoc Student–Newman–Keuls (SNK), Dunnett (in case of equal variance), or Dunnett’s T3 (in case of unequal variance) test. The value of *p* < 0.05 was considered statistically significant.

### 2.7. Ethical Approval

All procedures involving the handling of animals were conducted in accordance with the Chengdu University of Traditional Chinese Medicine Guide for the Care and Use of Laboratory Animals and were approved by the Institutional Ethics Committee of the Chengdu University of Traditional Chinese Medicine (protocol number 2018-21).

## 3. Results

### 3.1. Blood Pressure Measurement

To verify that high salt can cause salt-sensitive hypertension, we measured the systolic and diastolic blood pressure of rats. The basal level (before modeling) of blood pressure is shown in Appendix A. At the end of the modeling, the blood pressure result is shown in Figure 2, and the specific blood pressure values after modeling are shown in Appendix A. After 8 weeks of modeling, the blood pressure of the LSD and HSD groups significantly increased compared to the NC group. Furthermore, the blood pressure of the HSD group was also significantly higher than that of the LSD group.

This result suggests that DS rats are more likely to develop salt-sensitive hypertension than Wistar rats, and a high-salt diet further increases blood pressure. This conclusion is consistent with many previous research findings [14,15,16]. He et al. found that high salt not only increases hypertension but also affects endothelial cells and vascular structure [17]. This result may indicate that DS rats also experienced vascular endothelial injury.

### 3.2. Vascular Endothelial Injury Factor Determination

We detected the levels of vascular regulatory factors and inflammatory factors of the NC, LSD, and HSD groups according to the Elisa method. As shown in Figure 3, the levels of ET-1 and TNF-α in the HSD group were significantly higher than those in the LSD group. The level of Ang-ΙΙ in the NC group was significantly higher than that in the LSD and HSD groups. The levels of NO, IL-6, and IL-10 among the three groups were not significantly changed. The specific values of vascular endothelial injury factors are shown in Appendix A.

ET-1 has been widely studied as a potential risk marker for CVDs [18,19]. The elevated level of ET-1 indicates that high salt can damage endothelial function in the HSD group. The increase in Ang-ΙΙ level has been proven to promote endothelial cell proliferation and increase blood pressure [20]. However, the LSD and HSD groups with higher blood pressure had lower levels of Ang-ΙΙ. Maybe it suggests that the pathway for salt-sensitive hypertension to increase blood pressure is not through raising Ang-ΙΙ. The increased levels of TNF-α and IL-6 indicate that they cause inflammatory reactions [21], especially for TNF-α. The significant increase in TNF-α may be the main pathway of high-salt-induced inflammation. And IL-10 has an anti-inflammatory effect. In Xu’s study, the level of IL-10 has been shown to inhibit the production of many proinflammatory cytokines, including TNF-α and IL-6. And IL-10 also can prevent CVDs such as atherosclerosis and hypertension [22]. In this study, the levels of ET-1 and TNF-α significantly increased, which means that high salt would cause vascular endothelial injury and inflammatory reactions.

### 3.3. Artery Histopathological Observation

The artery pathological sections of 40 and 200 magnifications among the NC, LSD, and HSD groups are displayed in Figure 4. (Pathological sections at 100 magnification are shown in Appendix A). The diagrams in the NC group expressed a regular structure of the vascular wall without obvious endothelial cell damage. Comparatively, the LSD and HSD groups showed fibrosis and shrinkage of the vascular wall, and the structure of the endothelium was disrupted. It means salt-sensitive hypertension disrupts the endothelial structure of blood vessels. Between the LSD and HSD groups, the HSD group had higher blood pressure and more severe vascular endothelial injury. This result indicates that a high-salt diet not only aggravates hypertension but also exacerbates vascular damage.

### 3.4. Renal Histopathological Observation

The renal pathological sections of 100 and 200 magnifications among the NC, LSD, and HSD groups are displayed in Figure 5. In pathological sections of the NC group, no obvious damage was observed in the renal tubules and glomeruli. However, the LSD and HSD groups showed inflammatory infiltration, renal tubular necrosis, and glomerulosclerosis. The overall structure is unclear. This means that the DS rats showed renal injury. Especially in the HSD group, homogeneous protein-like substances could be seen in the renal tubules. This result indicates that compared to Wistar rats, DS rats are more prone to kidney damage.

### 3.5. Bile Acid Metabolomics Analysis

Based on the above results, compared to the NC group, the LSD and HSD groups showed elevated blood pressure and vascular endothelial injury. To explore the impact of salt intake on vascular endothelial injury, we used the LSD and HSD groups for further bile acid metabolomics analysis (Due to the different breeds of Wistar and DS rats, to avoid interfering with metabolomics results, the NC group was not used for further analysis).

According to bile acid metabolomics analysis, a total of thirty-eight bile acid compounds were successfully detected in the plasma samples. The example ion chromatography of plasma samples showed that various bile acids exhibited independent chromatographic peaks, indicating the good chromatographic separation of each bile acid (Figure 6A). Detailed information on all these bile acids is displayed in Appendix A. The enrichment analysis of these bile acids showed that bile acids are mainly related to primary bile acid biosynthesis, taurine and hypotaurine metabolism, and steroid hormone biosynthesis (Figure 6B). To verify the difference in bile acids between the LSD and HSD groups, the PLS-DA analysis was further used to explore the separation of bile acids in two groups. The result showed that the same group samples clustered together, and different group samples were well distinguished (Figure 6C). Among all these bile acids, the volcano plot showed that eleven types of bile acids had an obvious increase in the HSD group (Figure 6D).

Among the changed bile acids, based on VIP (variable important in projection) > 1, *p*-value < 0.05, and FC (Fold Change) > 2 or <0.5, glycocholic acid (GCA), taurolithocholic acid (TLCA), tauroursodeoxycholic acid (TUDCA), and glycolithocholic acid (GLCA) were identified as significantly differential bile acids between the LSD and HSD groups (Figure 7A and Appendix A). These results indicated that the high-salt diet caused obvious metabolic variations. To visualize the variation in significantly differential bile acids between the two groups, heat maps were plotted by MetaboAnalyst 5.0 (Figure 7B). The levels of GCA, TDCA, TUDCA, and GLCA increased in the HSD group, the group with more severe hypertensive vascular endothelial injury. This indicates that the four types of bile acids are all related to these pathological changes. Some studies can prove the connection between bile acids and vascular endothelial injury. Yntema et al. proposed that bile acids are the potential therapeutic targets of CVDs [23]. Chakraborty et al. found that some conjugated bile acids have anti-hypertensive effects [24]. The above conclusion indicates that bile acids can have a certain therapeutic effect on CVDs.

### 3.6. Correlation Analysis

In correlation analysis between vascular endothelial injury factors and bile acid levels, we obtained the correlation between the two results, as shown in Figure 8. The *p*-value for each comparison is shown in Appendix A. As shown in the result, the levels of ET-1 and TNF-α had a significantly positive correlation for all significantly differential bile acids. This means that the increase in these bile acids may be related to the increase in the levels of ET-1 and TNF-α. ET-1 and TNF-α will become the key factors in further research on the impact of bile acid metabolism on cardiovascular health. Cai et al. have proven that bile acids can regulate inflammation [25]. GCA and TLCA derivatives are naturally derived anti-inflammatory agents with high efficacy and safety [26,27]. Yao et al. found that GCA improved growth performance and alleviated hepatic cholestasis and tissue damage to the liver and intestine induced by a high-pectin diet [28]. Wu et al. found that TLCA and TUDCA rescue myelin phagocytosis in marrow-derived macrophages to reduce inflammation of the spinal cord [29]. Lee et al. found that the level of GLCA was significantly elevated after immunotherapy, which suggested that GLCA also has immune effects [30].

The above results suggest that bile acids may also reduce vascular endothelial injury and inflammation. The elevated levels of bile acids in the HSD group may indicate a protective mechanism against vascular endothelium injury and inflammation.

## 4. Discussion

In recent years, the incidence rate of CVDs has been increasing, and a high-salt diet is one of the important risk factors. However, the potential mechanism of the pathogenicity of high salt needs further study. Bile acids can regulate lipid metabolism and reduce the risk of atherosclerosis. Therefore, we believe there is also a correlation between bile acids and salt-sensitive hypertension-induced vascular endothelial injury.

The gene sequence of rats is similar to that of humans, and they have advantages such as strong disease resistance, gentle temperament, and easy breeding, making them particularly suitable for animal experiments and constructing disease models [31]. Wistar rats are one of the most common standardized experimental rat strains. In the cardiovascular disease model, DS rats have been cultivated based on Wistar rats [32]. Hypertension can be induced in DS rats by a high-salt diet, which has been widely used in animal experiments on hypertension and vascular disease [33]. In this study, DS rats were used to replicate animal models of salt-sensitive hypertension and vascular endothelial injury. Wistar rats were used as the normal control group of DS rats.

In the present study, after eight weeks of modeling, the blood pressure of the HSD group was significantly higher than that of the LSD group (*p* < 0.001). Meanwhile, under the same low-salt diet, the blood pressure of DS rats was significantly higher than that of Wistar rats (*p* < 0.01). Although the Ang-ΙΙ level in the NC group is higher, which is the reason for its elevated blood pressure [34], the LSD and HSD groups showed obvious renal injury, leading to more significant salt-sensitive hypertension [35]. In terms of vascular regulatory factors, the level of ET-1 of the HSD group is significantly higher than that of the LSD group (*p* < 0.01). ET-1 is one of the most widely known vascular regulators and can promote the fibrosis of vascular smooth muscle cells, induce inflammatory responses, and impair vascular endothelial function [36,37]. The increase in ET-1 levels and the result of pathological sections indicate high-salt-induced vascular endothelial injury [38,39]. Meanwhile, the level of inflammatory factor TNF-α in the HSD group significantly increased, which may mean that high-salt-induced inflammation is achieved by increasing the level of TNF-α. TNF-α is one of the cytokines involved in the acute phase of inflammation, participating in the systemic inflammatory response [40]. TNF-α can alter the characteristics of vascular endothelial cells and induce local coagulation [41]. The significant increase in TNF-α not only represents an inflammatory response in the HSD group but also proves the occurrence of blood stasis and vascular endothelial injury. The above results confirm that high salt can induce salt-sensitive hypertension, vascular endothelial injury, and inflammation, which provides evidence for the impact of high salt on the metabolic process and endocrine system.

In bile acid metabolomics analysis, enrichment analysis showed that most bile acids are involved in primary bile acid metabolism. Primary bile acids are further metabolized by intestinal microbiota in the intestine, ultimately forming secondary bile acids. Due to the ability of specific intestinal microbiota to express specific enzymes, the diversity of intestinal microbiota plays a crucial role in the generation of secondary bile acids [42]. Guo and Zhang found that the increase in bile acid levels under a high-salt diet reduces the number of *Akkermansia muciniphila* in rats’ feces, which is a beneficial bacteria negatively related to obesity, diabetes, CVDs, and inflammation [43,44]. In this study, four types of bile acids have been identified as significantly differential bile acids. These bile acids included GCA, TDCA, TUDCA, and GLCA, which are secondary bile acids. The levels of all secondary bile acids increased in the HSD group, which indicates high-salt-induced bile acid metabolism disorders and possible changes in intestinal microbiota.

In correlation analysis, the levels of ET-1 and TNF-α had a significantly positive correlation with all significantly differential bile acids. ET-1 is an important indicator of vascular endothelial injury, and TNF-α is closely related to inflammatory reaction and vascular endothelial injury [45,46]. This suggests that if the levels of ET-1 or TNF-α increase, it not only indicates vascular endothelial injury or inflammatory reactions but may also indicate a disruption of these bile acids. These results confirm that high-salt-induced vascular endothelial injury is related to the elevation of these bile acids.

This study identified bile acids associated with salt-sensitive hypertension-induced vascular endothelial injury and provided theoretical support for the prevention and treatment of CVDs. However, there are some shortcomings in this study. The direct evidence of the impact of bile acids on cardiovascular health is still lacking in systemic analysis. Bile acids are regulated by intestinal microbiota, and the changes in intestinal homeostasis and intestinal permeability that occur during salt-sensitive hypertension will be the next focus of study.

## Figures and Tables

**Figure 1 metabolites-14-00452-f001:**
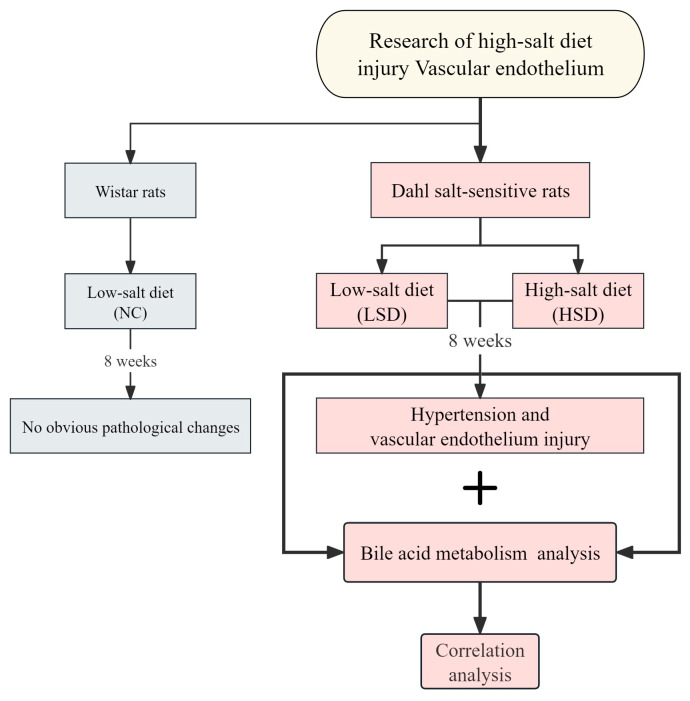
Schematic flowchart of this study.

**Figure 2 metabolites-14-00452-f002:**
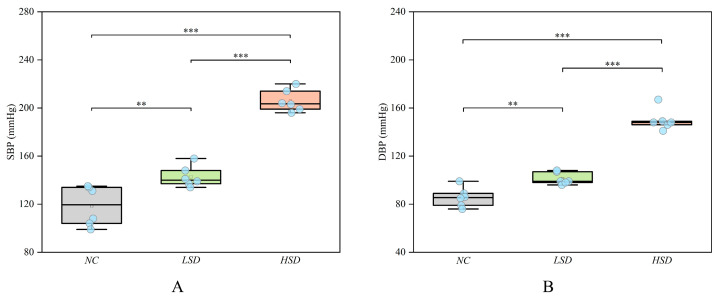
The blood pressure of the NC, LSD, and HSD groups in the 8th week. (**A**) The systolic pressure among the three groups. (**B**) The diastolic pressure among the three groups. ** *p* < 0.01, *** *p* < 0.001.

**Figure 3 metabolites-14-00452-f003:**
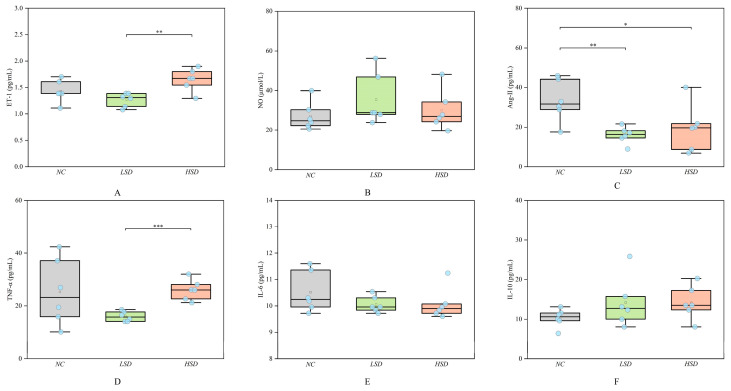
The levels of endothelial biomarkers and inflammatory factors of the NC, LSD, and HSD groups. (**A**) The level of ET-1. (**B**) The level of NO. (**C**) The level of ANG-ΙΙ. (**D**) The level of TNF-α. (**E**) The level of IL-6. (**F**) The level of IL-10. * *p* < 0.05, ** *p* < 0.01, *** *p* < 0.001.

**Figure 4 metabolites-14-00452-f004:**
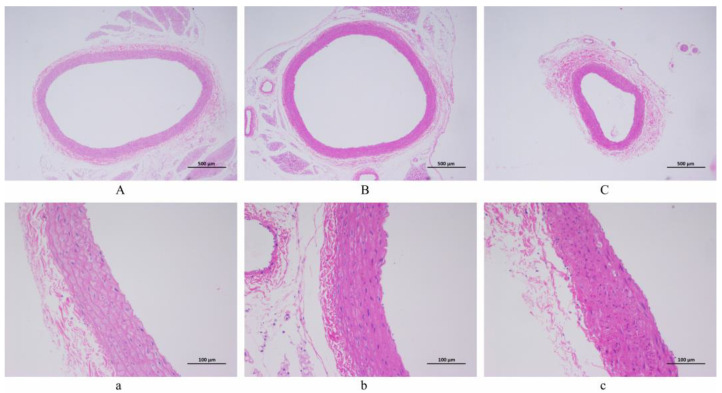
The histopathological observation of arterial blood vessels (H&E staining). (**A**,**a**) NC group. (**B**,**b**) LSD group. (**C**,**c**) HSD group. Scale bar: ((**A**–**C**) = 500 µm; (**a**–**c**) = 100 µm).

**Figure 5 metabolites-14-00452-f005:**
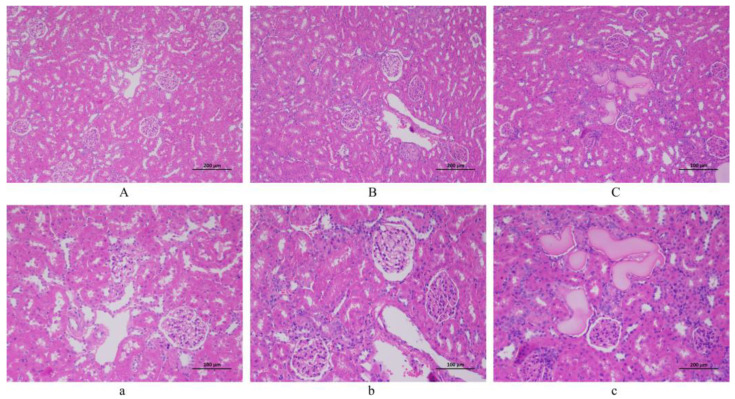
The histopathological observation of the kidney (H&E staining). (**A**,**a**) NC group. (**B**,**b**) LSD group. (**C**,**c**) HSD group. Scale bar: ((**A**–**C**) = 200 µm; (**a**–**c**) = 100 µm).

**Figure 6 metabolites-14-00452-f006:**
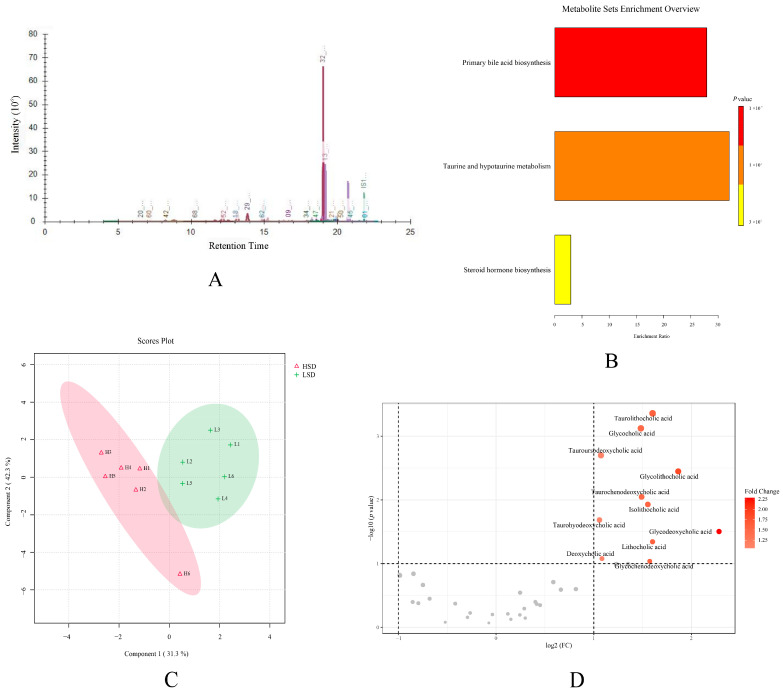
The results of bile acid metabolomics analysis. (**A**) The ion chromatography of the plasma sample. (**B**) The enrichment analysis of bile acids. (**C**) The PLS-DA score plot of the LSD and HSD groups. (**D**) The volcano plot between two groups.

**Figure 7 metabolites-14-00452-f007:**
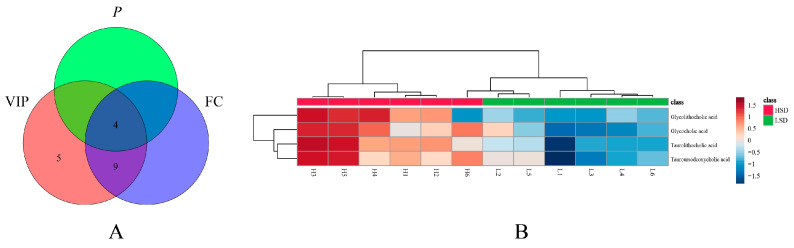
The results of differential bile acid screening. (**A**) The screening result of significantly differential bile acids. (**B**) The heat map of significantly differential bile acids between two groups.

**Figure 8 metabolites-14-00452-f008:**
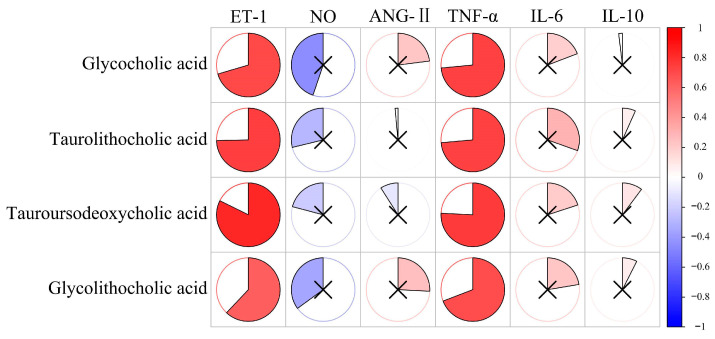
The correlation analysis between vascular endothelial injury factors and bile acids. The redder the color, the more positively correlated the two indicators. The bluer the color, the more negatively correlated the two indicators. × represents *p* > 0.05, with no significance.

## Data Availability

The original contributions presented in the study are included in the Appendix A; further inquiries can be directed to the first author.

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
