# Peer review of "Bile Acid Metabolism Analysis Provides Insights into Vascular Endothelial Injury in Salt-Sensitive Hypertensive Rats"

_metabolites, 2024, doi:10.3390/metabo14080452_

Round 1

Reviewer 1 Report

Comments and Suggestions for Authors

The authors analyzed the bile acid metabolism changes in rats with salt-sensitive hypertension-induced vascular endothelial injury. Original results were obtained that certainly deserve publication and will be of interest to a wide range of readers. However, some edits must be made to the text of the manuscript before publication.

Major concerns:

Lines 12-13: “We established a salt-sensitive hypertension-induced vascular endothelial injury rats model by detecting the levels of blood pressure, vascular regulatory factors, and inflammatory factors.” – the authors combined two sentences into one. The model was established using a high-salt diet and the success of this procedure was confirmed by detecting the levels of blood pressure, vascular regulatory factors, and inflammatory factors.

Line 14: “And evaluated pathological sections of arterial blood vessels.” – this is not a complete sentence.

Lines 21-22: “This study provided new evidence for the metabolic changes caused by salt-sensitive cardiovascular diseases.” – it is difficult to agree with this conclusion, since metabolic changes were caused not by salt-sensitive cardiovascular diseases but by high salt diet in rats with a genotype characterized as sensitive to a high salt diet.

Lines 44-46: As an important component of the metabolic process and endocrine system, bile acids are bound to have an impact on the occurrence and development of CVDs. – need to provide a link

Lines 50-54: problems with English

Figure 1: According to the figure, it can be understood that bile acid metabolism analysis was performed on LSD and HSD rats immediately after dividing the rats into groups. It is necessary to draw more correctly the arrows defining the steps of the experiment.

From the introduction it can be understood that the authors are the first to analyze the association of bile acid metabolism and vascular endothelial injury in salt-sensitive hypertension rats. However, the role of bile acids in cardiovascular health, including the function of vascular endothelial cells in salt-sensitive hypertension, was discussed earlier in a review article (Ishimwe JA, Dola T, Ertuglu LA, Kirabo A. Bile acids and salt-sensitive hypertension: a role of the gut-liver axis. Am J Physiol Heart Circ Physiol. 2022 Apr 1;322(4):H636-H646. doi: 10.1152/ajpheart.00027.2022.), which the authors do not refer to.

Lines 70-79: the text is given as a list of protocol steps. It is necessary to rewrite the text so that it reads as a description of methods, and not as an instruction to action.

Lines 83-84: same

Lines 151-155: Since blood pressure was measured only after 8 weeks of a low-salt or high-salt diet, it is incorrect to write “the blood pressure of the LSD and HSD groups significantly increased compared to the NC group”, since the authors do not know the basal level of pressure before the diet. Perhaps in LSD rats it did not change at all during the 8 weeks of the proposed diet. It is better to simply describe the results obtained.

Tables S1 and S2 and S4 – it is necessary to indicate what is shown in these tables: mean ± SEM?

Line 216: The enrichment analysis – not described in methods

Minor concerns:

Line 87: “EP tube” - needs to be decrypted

Line 101: UHPLC-PRM-MS – needs to be decrypted

Line 186: “study, The levels”

Line 212: “analysis, A total”

Line 216:” in Table. S3.”

Line 248: “Table. S5.”

Line 289: “Meanwhile, The level of”

Line 320: “The direct evidence on the impact of bile acids on cardiovascular health is still a lack of systemic analysis.” - check English, please

Throughout the text, spaces are required before square brackets with references cited.

Comments on the Quality of English Language

Line 14: “And evaluated pathological sections of arterial blood vessels.” – this is not a complete sentence.

Lines 50-54: problems with English

Lines 70-79: the text is given as a list of protocol steps. It is necessary to rewrite the text so that it reads as a description of methods, and not as an instruction to action.

Lines 83-84: same

Line 320: “The direct evidence on the impact of bile acids on cardiovascular health is still a lack of systemic analysis.” - check English, please

Reviewer 2 Report

Comments and Suggestions for Authors

Minor comments:

English grammar edits are necessary. Errors were found throughout the manuscript. Language can be improved.

Please remove the words - "kind of" in line 305

In the Methods section- The sample collection paragraph is written in the present tense, whereas the rest of the Methods section is in the past tense. Please have the methods in the past tense.

Please provide the catalog number of the rats' diet and the detailed composition in the supplementary files.

Please remove the words "DS rats can induce hypertension."- Instead, use "Hypertension can be induced in DS rats by high salt diet."

Major comments:

1.     Wistar rats(control) have higher Ang II but lower blood pressure compared to DS on both diets. Can the authors discuss this and provide the reasons?

2.     The authors did correlation analysis but did not test for causation, i.e., changes in bile acids induced ET and TNFα. Thus, they cannot say that ET-1 and TNF increases may also indicate disruption of bile acids.

3.       To improve the manuscript, this reviewer recommends the authors set up an endothelial cell culture model with and without endothelial cell bile acid receptors (FXR and TGR5) knockdown via siRNA, treat with any one or two of the differentially changed bile acids, and see if there is an increase in ET-1 and TNFα. This would show causation. 

Comments on the Quality of English Language

English grammar edits are necessary. Errors were found throughout the manuscript. Language can be improved.

In the Methods section- The sample collection paragraph is written in the present tense, whereas the rest of the Methods section is in the past tense. Please have the methods in the past tense

Round 2

Reviewer 1 Report

Comments and Suggestions for Authors

The authors have made corrections to the text of the manuscript according to the previously made comments. However, there are still a few minor comments that need to be taken into account before publication.

14-15: "And we evaluated the pathological sections of arterial blood vessels and kidney." is not a good phrase for an abstract. It would be better to write, for example, "The evaluation of the histological sections of arterial blood vessels and kidney confirmed the pathological processes in these tissues of experimental rats."

51: salt-sensitive hypertension rats – it would be more correct to write salt-sensitive hypertensive rats

 Methods: it is not described at what time point in the experiment and how the basal blood pressure was measured

Table S2: it is necessary to give a more detailed description of the data in the table, since it is not clear that these are the results of measuring the blood pressure of rats after a salt diet.

164-166: it is more logical to first write about measuring the basal blood pressure, and only then about measuring the blood pressure after a salt diet.

198: In this study, The levels

228: Figure 5. The Pathological section of the kidney – too brief and not entirely correct description of the figure, since the figure also shows the control, and in the control rats (according to the text) pathological processes in the kidneys were not found

Reviewer 2 Report

Comments and Suggestions for Authors

The authors have adequately addressed my major comments.

I  recommend edits to English. Please remove fragmented sentences like the examples shown below.

Abstract:

Line 13-15 "regulatory factors, and inflammatory factors. And we evaluated the pathological sections of arterial blood vessels and kidney.

Please make this a complete sentence without the "And we" fragmented sentence.

The following sentence in line 18 sounds abrupt and fragmented. "Meanwhile" is not necessary.

"Meanwhile, the high-salt diet causes disorders in bile acid metabolism."

Comments on the Quality of English Language

I recommend edits to English. Please remove fragmented sentences like the examples shown below.

Abstract:

Line 13-15 "regulatory factors, and inflammatory factors. And we evaluated the pathological sections of arterial blood vessels and kidney.

Please make this a complete sentence without the "And we" fragmented sentence.

The following sentence in line 18 sounds abrupt and fragmented. "Meanwhile" is not necessary.

"Meanwhile, the high-salt diet causes disorders in bile acid metabolism."
